# DELAYED GEOMETRIC DISCOUNTS: AN ALTERNATIVE CRITERION FOR REINFORCEMENT LEARNING

## ABSTRACT

The endeavor of artificial intelligence (AI) is to design autonomous agents capable of achieving complex tasks. Namely, reinforcement learning (RL) proposes a theoretical background to learn optimal behaviors. In practice, RL algorithms rely on geometric discounts to evaluate this optimality. Unfortunately, this does not cover decision processes where future returns are not exponentially less valuable. Depending on the problem, this limitation induces sample-inefficiency (as feed-backs are exponentially decayed) and requires additional curricula/exploration mechanisms (to deal with sparse, deceptive or adversarial rewards). In this paper, we tackle these issues by generalizing the discounted problem formulation with a family of delayed objective functions. We investigate the underlying RL problem to derive: 1) the optimal stationary solution and 2) an approximation of the optimal non-stationary control. The devised algorithms solved hard exploration problems on tabular environment and improved sample-efficiency on classic simulated robotics benchmarks.

## 1 INTRODUCTION

In the infinite horizon setting, and without further assumptions on the underlying Markov Decision Process (MDP), available RL algorithms learn optimal policies only in the sense of the discounted cumulative rewards (Puterman, 2014). While the geometric discounting is well suited to model a termination probability or an exponentially decaying interest in the future, it is not flexible enough to model alternative weighting of the returns. Consider for example settings where the agent is willing to sacrifice short term rewards in favor of the long term outcome. Clearly, for such situations, a discounted optimality criterion is limited and does not describe the actual objective function.

This is particularly true in hard exploration tasks, where rewards can be sparse, deceptive or adversarial. In such scenarios, performing random exploration can rarely lead to successful states and thus rarely obtain meaningful feedback. Geometric discounts tend to fail in such scenarios. Consider for example the U-maze environment in Figure 1a, where the reinforcement signal provides a high reward ($+1$) when reaching the green dot in the bottom arm, a deceptive reward ($+0.9$) when reaching the blue dot in the upper arm, and a negative reward ($-1$) for crossing the red corridor. If the agent is only interested in the long term returns, then the optimal control should always lead to the green dot. However, depending on the initial state, optimal policies in the sense of the discounted RL problem are likely to prefer the deceptive reward due to the exponentially decaying interest in the future (Figure 1b).

Naturally, higher discount factors are associated with optimal policies that also optimize the average returns (Blackwell, 1962), which can solve in principle the described hard exploration problem. However, in practice, such discount values can be arbitrarily close to 1 which entails severe computational instabilities. In addition, and particularly in continuous settings or when tasks span over extremely long episodes, discount-based RL approaches are sample-inefficient and are slow at propagating interesting feed-backs to early states.

In this paper, we generalize the geometric discount to derive a variety of alternative time weighting distributions and we investigate the underlying implications of solving the associated RL problem both theoretically and practically. Our contributions are twofold. First, we introduce a novel family that generalizes the geometrically discounted criteria, which we call the *delayed discounted criteria*. Second, we derive tractable solutions for optimal control for both stationary and non-stationary

policies using these novel criteria. Finally, we evaluate our methods on both hard exploration tabular environments and continuous long-episodic robotics tasks where we show that:

1. Our agents can solve the hard exploration problem in a proof of concept setup.
2. Our methods improve sample-efficiency on continuous robotics tasks compared to Soft-Actor-Critic.

Figure 1c showcases how non-geometrically discounted criteria impacts the profile of optimal value function in the U-maze example.

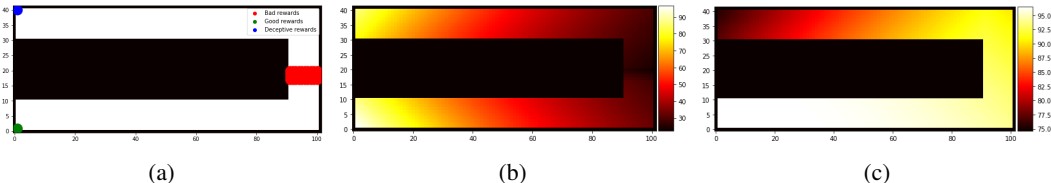

(a)                (b)                (c)

Figure 1: (a) Hard exploration problem example, (b) optimal value function of geometrically discounted RL with a discount factor $\gamma = 0.99$ and (c) non-geometrically discounted RL.

## 2   REINFORCEMENT LEARNING WITH NON GEOMETRIC DISCOUNT

Consider an infinite horizon Markov Decision Process (MDP) $\mathcal{M} = \{\mathcal{S}, \mathcal{A}, \mathcal{P}, r, \gamma, p_0\}$, where $\mathcal{S}$ and $\mathcal{A}$ are either finite or compact subsets of respectively $\mathbb{R}^d$ and $\mathbb{R}^{d'}$, $(d, d') \in \mathbb{N}^2$, $\mathcal{P} : \mathcal{S} \times \mathcal{A} \to \Delta(\mathcal{S})$ is a state transition kernel[1], $r : \mathcal{S} \times \mathcal{A} \to \mathbb{R}$ is a continuous reward function, $p_0 \in \Delta(\mathcal{S})$ an initial state distribution and $\gamma \in (0, 1)$ is the discount factor. A policy $\pi$ is a mapping indicating, at each time step $t \in \mathbb{N}$, the action $a_t$ to be chosen at the current state $s_t$. The goal of geometrically discounted reinforcement learning algorithms is to optimize the discounted returns:

$$\mathcal{L}(\pi, r) := \mathbb{E}_{\pi, p_0}\Big[ \sum_{t=0}^{\infty} \gamma^t r_t \Big] \quad ; \quad r_t := r(s_t, a_t)$$

where $\mathbb{E}_{\pi, p_0}$ denotes the expectation over trajectories generated in $\mathcal{M}$ using the policy $\pi$ and initialised according to $p_0$.

In this section, we present our methods. First, Section 2.1 introduces our delayed discounted family of optimality criteria. Then, Section 2.2 investigates the optimization of the linear combination of this new family in the context of stationary policies. Finally, Section 2.3 generalizes the optimal control to non-stationary policies.

### 2.1   BEYOND THE GEOMETRIC DISCOUNT: A DELAYED DISCOUNTED CRITERION

We propose to investigate a particular parametric family of optimality criteria that is defined by a sequence of discount factors. For a given delay parameter $D \in \mathbb{N}$, we define the discount factors $\gamma_d \in (0, 1)$ for any integer $d \in [0, D]$ and we consider the following loss function:

$$\mathcal{L}_D(\pi, r) := \mathbb{E}_{\pi, p_0}\Big[ \sum_{t=0}^{\infty} \Phi_D(t) r_t \Big] \quad where \quad \Phi_D(t) := \sum_{\substack{\{a_d \in \mathbb{N}\}_{d=0}^{D} \\ such\ that\ \sum_d a_d = t}} \prod_{d=0}^{N} \gamma_d^{a_d} \qquad (1)$$

This class of optimality criteria can be seen as a generalization of the classical geometric discount. In fact, we highlight that for $D = 0$, $\Phi_0(t) = \gamma_0^t$ which implies that $\mathcal{L}_0(\pi, r) = \mathcal{L}(\pi, r)$ for any policy $\pi$ and any reward function $r$. In Figure 2, we report the normalized distribution of the weights $\Phi_D(t)$ (i.e. $y(t) = \frac{\Phi_D(t)}{\sum_i \Phi_D(i)}$) over time as we vary the parameter $D \in \{0, \ldots, 9\}$. Notice how the mode of the probability distribution is shifted towards the right as we increased the delay, thus putting more weights on future time steps.

---

[1]$\Delta(\mathcal{S})$ denotes the set of probability measures over $\mathcal{S}$

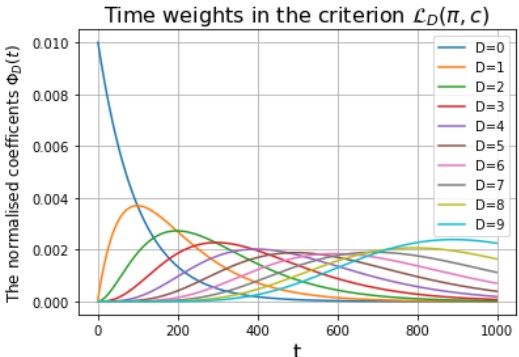

Figure 2: The normalized coefficients $\Phi_D(t)$ over time for different values of the delay parameter $D$

Intuitively, the proposed criterion ($\mathcal{L}_D$) describes the goal of an agent that discards short term gains in favor of long term discounted returns. In the following, we consider yet a more diverse problem formulation by using a linear combination of the delayed losses. Let $\mathcal{L}_\eta$ be the following objective function defined as:

$$\mathcal{L}_\eta(\pi, r) := \mathbb{E}_{\pi, p_0}\left[\sum_t \eta(t) r_t\right] \quad \text{such that} \quad \eta(t) = \sum_{d=0}^{D} w_d \Phi_d(t) \tag{2}$$

where the depth $D \in \mathbb{N}$ and the coefficients $w_d \in \mathbb{R}$ for any $d \in [0, D]$ are known.

In general, the optimal control in the sense of $\mathcal{L}_\eta$ is not stationary. However, learning solutions that admit compact representations is crucial for obvious computational reasons. In the following we propose to learn either the optimal stationary solution, or to approximate the optimal control with a non-stationary policy over a finite horizon followed by a stationary one.

## 2.2 Optimal stationary policies

In this section, we propose to learn stationary solution using a policy-iteration like algorithmic scheme. As in the classical setting, the goal is to learn a control $\pi^*$ that maximizes the state-action value function $Q_\eta^\pi$:

$$Q_\eta^\pi(s, a) = \sum_{d=0}^{D} w_d Q_d^\pi(s, a) \quad \text{where} \quad Q_d^\pi(s, a) := \mathbb{E}_\pi\left[\sum_{t=0}^{\infty} \Phi_d(t) r_t | s_0, a_0 = s, a\right] \tag{3}$$

This is done by iteratively evaluating $Q_\eta^\pi$ and then updating the policy to maximize the learned value function. Due to the linearity illustrated in Equation 3, the policy evaluation step is reduced to the estimation of $Q_d^\pi$. In geometrically discounted setting, the value function is the fixed point of the Bellman optimality operator. Luckily, this property is also valid for the quantities $Q_d^\pi$:

**Proposition 1** *For any discount parameters $(\gamma_d)_{d=0}^{D}$, the value functions $Q_D^\pi$ is the unique fixed point of the following $\gamma_D$-contraction:*

$$[T_\pi^D(q)](s, a) = \underbrace{\mathbb{E}_{\substack{s' \sim \mathcal{P}(s,a) \\ a' \sim \pi(s')}}\left[r(s, a) + \sum_{d=0}^{D-1} \gamma_d Q_d^\pi(s', a')\right]}_{:= r_D^\pi(s, a)} + \gamma_D \mathbb{E}_{\substack{s' \sim \mathcal{P}(s,a) \\ a' \sim \pi(s')}}\left[q(s', a')\right]. \tag{4}$$

The value $Q_D^\pi$ of a policy $\pi$ with respect to the delayed criterion $\mathcal{L}_D$ is the state-action value function of the same policy using an augmented policy dependent reward $r_D^\pi$ w.r.t. the $\gamma_D$-discounted returns.

Intuitively, the instantaneous worth of an action ($r_D^\pi$) is the sum of the environments' myopic returns ($r(s, a)$) and the long term evaluations (with lower delay parameters ($Q_d^\pi$)$_{d<D}$).

This has the beneficial side-effect of enhancing sample efficiency as it helps the agent to rapidly back-propagate long-term feed-backs to early states. This reduces the time needed to distinguish good from bad behaviors, particularly in continuous settings where function approximation are typically used to learn policies. This is discussed in details in Section 4.2.

Similarly to standard value based RL algorithms, given a data set of trajectories $\mathcal{D} = \{s, a, s'\}$, the value function $Q_d^\pi$ can be approximated with parametric approximator $Q_{\theta_d}$ by optimising $J_d^Q(\theta)$:

$$J_D^Q(\theta) = \mathbb{E}_{\substack{s,a,s' \sim \mathcal{D} \\ a' \sim \pi_\phi(s')}} \Big[ \frac{1}{2} \big( Q_\theta - (r(s,a) + \sum_{d=0}^{D} \gamma_d Q_{\bar{\theta}_d}(s',a'))^2 \big) \Big] \tag{5}$$

As for the policy update step, inspired from the Soft-Actor-Critic (SAC) algorithm, we propose to optimize an entropy regularized soft Q-value using the following loss where $\alpha$ is a parameter:

$$J_\eta^\pi(\phi) = -\mathbb{E}_{s \sim \mathcal{D}, a \sim \pi_\phi} \Big[ \sum_{d=0}^{D} w_d Q_{\bar{\theta}_d}(s,a) - \alpha \log(\pi_\phi(a|s)) \Big] \tag{6}$$

We use Equations 5 and 6 to construct Algorithm 1, a generalization of the SAC algorithm that approximates optimal stationary policies in the sense of $\mathcal{L}_\eta$. In practice, this can be further improved using the double Q-network trick and the automatic tuning of the regularization parameter $\alpha$. This is discussed in Appendix A.1. Unfortunately, unlike the geometrically discounted setting, the policy improvement theorem is no longer guaranteed in the sense of $\mathcal{L}_\eta$. This means that depending on the initialization parameters, Algorithm 1 can either converge to the optimal stationary control or get stuck in a loop of sub-optimal policies. This is discussed in detail in Section 4.1.

---

**Algorithm 1** Generalized Soft Actor Critic

1: **Input:** initial parameters $(\theta_d)_{d=0}^{D}, \phi$, learning rates $(\lambda_d)_{d=0}^{D}, \lambda_\pi$, pollyak parameter $\tau$
2: initialise target network $\bar{\theta}_d \leftarrow \theta_d$ and initialise replay buffer $\mathcal{D} \leftarrow \emptyset$
3: **for** each iteration **do**
4:     **for** each environment step **do**
5:         $a_t \sim \pi_\phi(s_t)$, $s_{t+1} \sim \mathcal{P}(s_t, a_t)$, $\mathcal{D} \leftarrow \mathcal{D} \cup \{(s_t, a_t, s_{t+1}, c_t)\}$
6:     **for** $d \in [0, D]$ **do**
7:         **for** each $Q_d$ gradient step **do**
8:             update parameter $\theta_d \leftarrow \theta_d - \lambda_d \hat{\nabla}_{\theta_d} J_d^Q(\theta_d)$ and update target $\bar{\theta}_d \leftarrow \tau \bar{\theta}_d + (1 - \tau)\theta_d$
9:     **for** each policy update **do**
10:         update policy $\phi \leftarrow \phi - \lambda_\pi \hat{\nabla}_\phi J_\eta^\pi(\phi)$
11: **Return:** $\pi_\phi, (Q_{\bar{\theta}_d})_{d=0}^{D}$

---

### 2.3 APPROXIMATE OPTIMAL CONTROL

In the general case, the optimal control is not necessarily stationary. Consider the problem of learning the optimal action profile $a_{0:\infty}^*$ which yields the following optimal value function:

$$V_\eta^*(s) := \max_{a_{0:\infty}} \mathbb{E}_{a_{0:\infty}} \Big[ \sum_{t=0}^{\infty} \eta(t) r_t | s_0 = s \Big], \tag{7}$$

where $\mathbb{E}_{a_{0:\infty}}$ is the expectation over trajectories generated using the designated action profile. In this section we propose to solve this problem by approximating the value function from Equation 7. This is achieved by applying the Bellman optimality principle in order to decompose $V_\eta^*$ into a finite horizon control problem (optimising $a_{0:H}$ with $H \in \mathbb{N}$) and the optimal value function $V_{f^{H+1}(\eta)}^*$

where $f$ is an operator transforming the weighting distribution as follows:

$$\Big[f(\eta)\Big](t) := \sum_{d=0}^{D} \Big(\gamma_d \sum_{j=d}^{D} w_j\Big) \Phi_d(t) = \langle \Gamma \cdot \mathbf{w}, \mathbf{\Phi}(t) \rangle$$

$$where \quad \Gamma := \begin{bmatrix} \gamma_0 & \gamma_0 & \gamma_0 & \cdots & \gamma_0 \\ 0 & \gamma_1 & \gamma_1 & \cdots & \gamma_1 \\ 0 & 0 & \gamma_2 & \cdots & \gamma_2 \\ \vdots & & & \ddots & \vdots \\ 0 & 0 & 0 & \cdots & \gamma_D \end{bmatrix} \quad ; \quad \mathbf{w} := \begin{bmatrix} w_0 \\ w_1 \\ w_2 \\ \vdots \\ w_D \end{bmatrix} \quad ; \quad \mathbf{\Phi}(t) := \begin{bmatrix} \Phi_0(t) \\ \Phi_1(t) \\ \Phi_2(t) \\ \vdots \\ \Phi_D(t) \end{bmatrix}$$

For the sake of simplicity, let $\langle \mathbf{1}, \eta \rangle := \sum_{d=0}^{D} w_d$ and $\big[f^n(\eta)\big](t) := \langle \Gamma^n \cdot \mathbf{w}, \mathbf{\Phi}(t) \rangle$ for any $n, t \in \mathbb{N}$.

**Proposition 2** *For any state $s_0 \in \mathcal{S}$, the following identity holds:*

$$V_\eta^*(s_0) = \max_{a_{0:H}} \Big\{ \mathbb{E}_{a_{0:H}} \Big[ \sum_{t=0}^{H} \langle \mathbf{1}, f^t(\eta) \rangle r_t \Big] + \mathbb{E}_{a_0, a_1, .., a_H} \big[ V_{f^{H+1}(\eta)}^*(s_{H+1}) \big] \Big\} \tag{8}$$

As a consequence, the optimal policy in the sense of $\mathcal{L}_\eta$ is to execute the minimising arguments $a_{0:H}^*$ of Equation 8 and the execute the optimal policy in the sense $\mathcal{L}_{f^{H+1}(\eta)}$. Unfortunately, solving $\mathcal{L}_{f^{H+1}(\eta)}$ is not easier than the original problem.

However, under mild assumptions, for $H$ large enough, this criterion can be approximated with a simpler one. In order to derive this approximation, recall that the power iteration algorithm described by the recurrence $\mathbf{v}_{k+1} = \frac{\Gamma \cdot \mathbf{v}_k}{\|\Gamma \cdot \mathbf{v}_k\|}$ converges to the unit eigenvector corresponding to the largest eigenvalue of the matrix $\Gamma$ whenever that it is diagonalizable. In particular, if $0 < \gamma_D < ... < \gamma_0 < 1$ and $\mathbf{v}_0 = \mathbf{w}$, then $\Gamma$ is diagonalizable, $\gamma_0$ is it's largest eigenvalue and the following holds:

$$\lim_{n \to \infty} \mathbf{v}_{n+1} = \lim_{n \to \infty} \frac{\Gamma^n \cdot \mathbf{w}}{\prod_{k \leq n} \|\Gamma \cdot \mathbf{v}_k\|} = [\mathbb{1}_{i=0}]_{i \in [0, D]} \quad and \quad \lim_{n \to \infty} \frac{V_{f^n(\eta)}^*(s)}{\prod_{k \leq n} \|\Gamma \cdot \mathbf{v}_k\|} = V^*(s) \tag{9}$$

Under these premises, the right hand term of the minimisation problem in Equation 8 can be approximated with the optimal value function in the sense of the classical RL criterion $\mathcal{L}$ (which optimal policy can be computed using any standard reinforcement learning algorithm in the literature).

Formally, we propose to approximate Equation 8 with a proxy optimal value function $\tilde{V}_{\eta,H}^*(s_0)$:

$$\tilde{V}_{\eta,H}^*(s_0) = \max_{a_{0:H}} \Big\{ \mathbb{E}_{a_{0:H}} \Big[ \sum_{t=0}^{H} \langle \mathbf{1}, f^t(\eta) \rangle r_t \Big] + \Big( \prod_{k \leq H} \|\Gamma \cdot \mathbf{v}_k\| \Big) \mathbb{E}_{a_0, a_1, .., a_H} \big[ V^*(s_{H+1}) \big] \Big\} \tag{10}$$

A direct consequence of Equation 9 is that $\lim_{H \to \infty} \tilde{V}_{\eta,H}^*(s) = V_\eta^*(s)$ for any state $s \in \mathcal{S}$. In addition, for a given horizon $H$, the optimal decisions in the sense of the proxy problem formulation are to execute for the first $H$ steps the minimising arguments $a_{0:H}^*$ of Equation 10 (they can be computed using dynamic programming) and then execute the optimal policy in the sense of the $\gamma_0$-discounted RL (which can be computed using any standard RL algorithm).

---

**Algorithm 2** H-close optimal control

---

1: Compute $\pi^*, V^* \leftarrow$ POLICY ITERATION and initialize $\boldsymbol{v}_0 \leftarrow \boldsymbol{w}$
2: **for** $t \in [1, H]$ **do**
3:      Compute $\boldsymbol{v}_t \leftarrow \Gamma \cdot \boldsymbol{v}_{t-1}$
4: Initialise $\boldsymbol{V}_{H+1}(s) \leftarrow V^*(s) \cdot \prod_{k \leq H} \|\Gamma \cdot \mathbf{v}_k\|$
5: **for** $t \in [H, 0]$ **do**
6:      Solve $\pi_t(s) \leftarrow \arg\max_a \|\boldsymbol{v}_t\| \cdot c(s, a) + \mathbb{E}_{s' \sim \mathcal{P}(s,a)} \big[ \boldsymbol{V}_{t+1}(s') \big]$
7:      Compute $\boldsymbol{V}_t(s) \leftarrow \|\boldsymbol{v}_t\| \cdot c(s, \pi_t(s)) + \mathbb{E}_{s' \sim \mathcal{P}(s, \pi_t(s))} \big[ \boldsymbol{V}_{t+1}(s') \big]$
8: **Return:** $(\pi_t)_{t \in [0, H]}, \pi^*$

---

## 3 RELATED WORK

**Bridging the gap between discounted and average rewards criteria.** It is well known that defining optimality with respect to the cumulative discounted reward criterion induces a built-in bias against policies with longer mixing times. In fact, due to the exponential decay of future returns, the contribution of the behaviour from the $T^{th}$ observation up to infinity is scaled down by a factor of the order of $\gamma^T$. In the literature, the standard approach to avoid this downfall is to define optimality with respect to the average reward criterion $\bar{\mathcal{L}}$ defined as :

$$\bar{\mathcal{L}}(\pi, c) := \lim_{T \to \infty} \frac{1}{T} \mathbb{E}_{\pi, p_0} \Big[ \sum_{t=0}^{T} r_t \Big]$$

This setting as well as dynamic programming algorithms for finding the optimal average return policies have been long studied in the literature (Howard, 1960; Veinott, 1966; Blackwell, 1962; Puterman, 2014). Several value based approaches (Schwartz, 1993; Abounadi et al., 2001; Wei et al., 2020) as well as policy based ones (Kakade, 2001; Baxter & Bartlett, 2001) have been investigated to solve this problem. These approaches are limited in the sense that they require particular MDP structures to enjoy theoretical guarantees.

Another line of research, is based on the existence of a critical discount factor $\gamma_{crit} < 1$ such that for any discount $\gamma \in (\gamma_{crit}, 1)$ the optimal policy in the sense of the $\gamma$-discounted criterion also optimises the average returns Blackwell (1962). Unfortunately, this critical value can be arbitrarily close to 1 which induces computational instabilities in practice. For this reason, previous works attempted to mitigate this issue by increasing the discount factor during training (Prokhorov & Wunsch, 1997), learning higher-discount solution via learning a sequence of lower-discount value functions (Romoff et al., 2019) or tweaking the reinforcement signal to equivalently learn the optimal policy using lower discounts (Tessler & Mannor, 2020).

**Exploration Strategies.** Another line of research attempted to tackle the hard exploration problem by further driving the agents exploration towards interesting states. Inspired by intrinsic motivation in psychology (Oudeyer & Kaplan, 2008), some approaches train policies with rewards composed of extrinsic and intrinsic terms. Namely, count-based exploration methods keep track of the agents' past experience and aim at guiding them towards rarely visited states rather than common ones (Bellemare et al., 2016; Colas et al., 2019). Alternatively, prediction-based exploration define the intrinsic rewards with respect to the agents' familiarity with their environments by estimating a dynamics prediction model (Stadie et al., 2015; Pathak et al., 2019). Other approaches maintain a memory of interesting states (Ecoffet et al., 2019; 2021), trajectories (Guo et al., 2019) or goals (Guo & Brunskill, 2019). Ecoffet et al. (2019; 2021) first return to interesting states using either a deterministic simulator or a goal-conditioned policy and start exploration from there. (Guo et al., 2019) train a trajectory-based policy to rather prefer trajectories that end with rare states. Guo & Brunskill (2019) revisit goals that have higher uncertainty. Finally, based on the options framework (Sutton et al., 1999), options-based exploration aims at learning policies with termination conditions—or macro-actions. This allows the introduction of abstract actions, hence driving the agents' exploration towards behaviours of interest (Gregor et al., 2016; Achiam et al., 2017).

## 4 EXPERIMENTS

In this section, we evaluate the performance of the proposed algorithm. Our goal is to answer the following questions:

- How does the different parameters impact the ability of the proposed algorithms to solve hard exploration problems?
- How does the proposed algorithm impact the performance in classical continuous control problems?

### 4.1 HARD EXPLORATION PROBLEMS

In this section we investigate our ability to approximate the solution of the proposed family of RL problems (w.r.t. the optimality criterion $\mathcal{L}_D$) in discrete hard exploration maze environments. We

fix the discount factor values to $\gamma_i = 0.99 - \frac{i}{1000}$ throughout the experiments. This guarantees that $\Gamma$ is diagonalizable and that its largest eigenvalue is $\gamma_0 = 0.99$. We evaluate the performance of both Algorithms 1 and 2 as we vary the depth parameter and as we increase the non stationarity horizon.

In Figure 3 we reported the shapes of the mazes as well as the used reinforcement signal in each of them. We used a sparse signal where the green dots represent the best achievable reward, the blue dots are associated with a deceptive reward and the red dots are associated with a penalty. This setting is akin to hard exploration problems as the agent might learn sub-optimal behaviour because of the deceiving signal or because of the penalty.

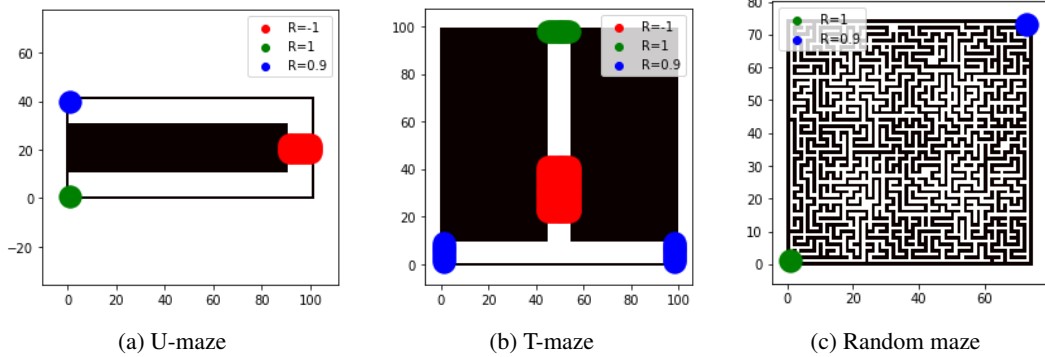

|        (a) U-maze        |        (b) T-maze        |        (c) Random maze        |

Figure 3: Hard exploration environments

**Stationary solutions:**   We start by evaluating the performances of the learned policies using Generalised Soft Actor Critic (GSAC) (Algorithm 1). As discussed earlier, unlike the geometrically discounted setting, the policy update is not guaranteed to improve the performances. For this reason, depending on the initialisation, the algorithm can either converge to the optimal stationary policy, or get stuck in a sequence of sub-optimal policies.

In Figure 4, we reported the performances of GSAC as we varied the depth hyper-parameter $D$ using two initialisation protocol. The blue curves are associated with a randomly selected initial parameters while the red curves are associated with experiment were the policy is initialised with the solution of the geometrically discounted problem. The solid lines correspond to the average reward while the dashed lines correspond to the $\Phi_d$ weighted loss (as computed in $\mathcal{L}_d$).

In all experiments, the expectations were averaged across 25 runs of the algorithm using trajectories of length 4000 initialised in all possible states (uniform $p_0$) . A common observation is that for a depth $D$ higher than 6 7 the algorithm was unstable and we couldn't learn a good stationary policy in a reliable way. However, the learned stationary policies with even a relatively shallow depth parameter yielded reliable policies that not only maximise the $\Phi_D$ weighted rewards but also improved the average returns. Notice how the baseline ($D = 0$, i.e. the geometrically discounted case) always under-performs when compared to the learned policies for a depth parameter around 3.

We also observe that the algorithm is sensitive to the used initialisation. Using the optimal policy in the sens of the geometrically discounted objective (red curves) helped stabilise the learning procedure in most cases: this is particularly true in the random maze environment where a random initialisation of the policy yielded bad performances even with a low depth parameter. This heuristic is not guaranteed to produce better performances in all cases, notice how for $D = 7$ in the T-maze, a random initialisation outperformed this heuristic consistently.

**Optimal Control Approximation:**   In this section we investigate the performances of the policies learned using Algorithm 2 as we vary the non stationarity horizon $H$ for three depth parameters $D \in \{5, 10, 15\}$ (respectively the red, blue and green curves). As in the last experiment, we reported the expected returns as the average rewards using the continuous lines and as the $\Phi_D$ weighted rewards using dashed lines. As a baseline, we can observe in each figure the performances of the optimal policy in the sense of the geometrically discounted criterion when the non-stationarity

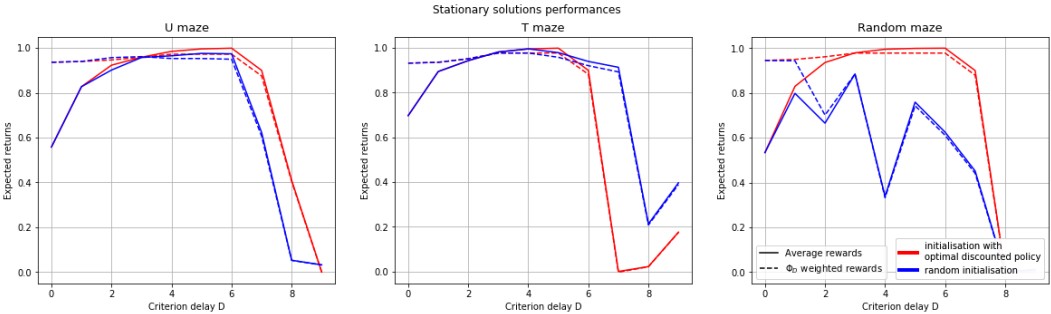

Figure 4: Learned stationary policy (GSAC) performances as the depth parameter varies

horizon $H = 0$. We also reported the performances of the best stationary policy learned using GSAC for a depth parameter $D = 5$.

The first notable observation is that increasing the hyper-parameter $H$ does indeed improve performances. In addition, we notice that the maximum possible improvement is reached using a finite horizon $H$ (i.e. the maximising argument of both $\tilde{V}_{\eta,H}$ and $V_\eta^*$ are the same). Intuitively, the $H$ non stationary steps enable the agent to get to an intermediate state from which the optimal policy in the sense of the discounted RL formulation can lead to the state with the highest rewards. This explains the effectiveness of current hard exploration RL algorithmsEcoffet et al. (2019); Eysenbach et al. (2019): by learning a policy that reaches interesting intermediate state, these methods are implicitly learning an approximate solution of $\tilde{V}_{\eta,H}$.

In the case of $D = 5$, the obtained performances using Algorithm 2 converged to the performances of the optimal stationary policy obtained using GSAC in both the random and T-maze. On the other hand, unlike GSAC, the H-close algorithm is not sensitive to initialisation: in fact notice how even for arbitrarily high depth parameter ($D = 15$) the learned policy ends up saturating the average and the $\Phi_D$ weighted rewards. More interestingly, increasing the depth can be beneficial as we observe empirically that for higher depth parameters, the non-stationarity horizon required to achieve the best possible performances decreases.

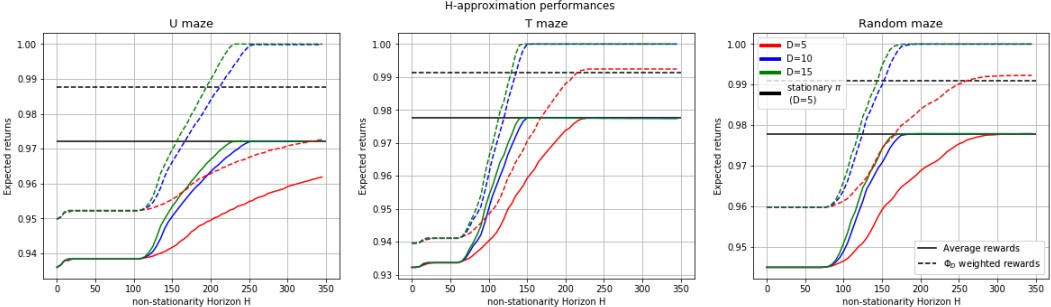

Figure 5: Learned H-close optimal control

## 4.2 SIMULATED CONTINUOUS CONTROL BENCHMARKS

In this section, we propose to evaluate our methods on several continuous robotics domains. To this end, we consider 5 different environments within the MUJOCO physics engine (Todorov et al., 2012): Walker2d, Hopper, HalfCheetah, Ant and Humanoid. As results in the tabular settings showed that there exists a threshold beyond which increasing the value of the delay $D$ hurts the performance, we fix $D = 1$, where only two discount factors $\gamma_0$ and $\gamma_1$ are used. We compare our proposed methods to the classic SAC algorithm to investigate the implications of using both critics $Q_{\theta_0}$ and $Q_{\theta_1}$.

In Figure 6, we report the average rewards obtained by the different agents over time. Note that in all the environments, the GSAC agents are faster in collecting positive rewards than the SAC

agents. The main reason behind this is that the critic tends to discern good from bad actions in the GSAC agents faster than the SAC agents. In fact, on the one hand, the SAC agents spend more time uncertain about the quality of their actions for a given state, and thus need more time and more experience to make the estimations of their critic more accurate and reflecting the true long-term value of an action. On the other hand, in the GSAC agents, $Q_{\theta_1}$ takes into consideration the estimate of the actions by $Q_{\theta_0}$, and thus there is less uncertainty about their quality.

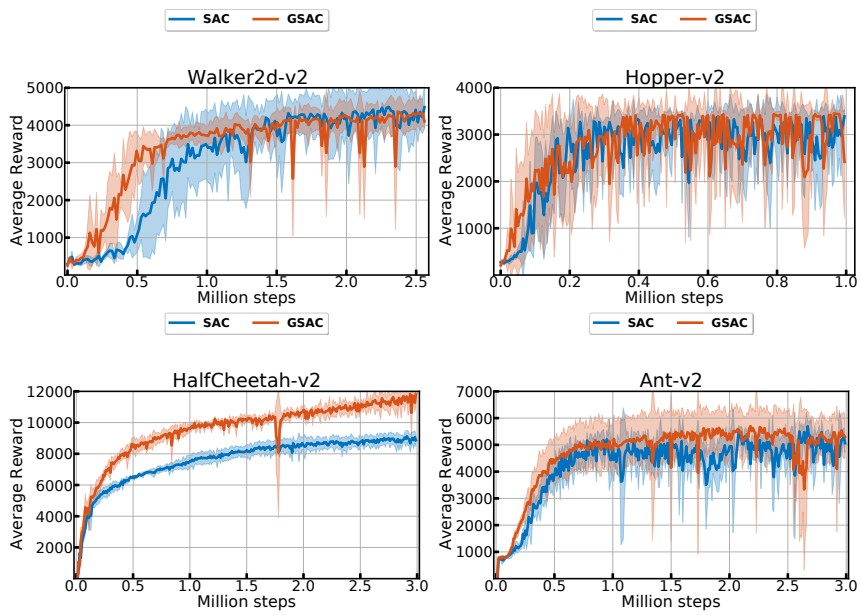

Figure 6: Training curves on continuous control benchmarks. Generalized Soft-Actor-Critic (GSAC) shows better sample efficiency across all tasks

## 5 CONCLUSION

Designing human-like autonomous agents requires (among other things) the ability to discard short term returns in favor of long term outcomes. Unfortunately, existing formulations of the reinforcement learning problem stand on the premise of either discounted or average returns: both providing a monotonic weighting of the rewards over time.

In this work, we propose a family of delayed discounted objective functions that captures a wide range of non-monotonic time-preference models. We analyse the new formulation to construct 1) the Bellman optimality criterion of stationary solution and 2) a feasible iterative scheme to approximate the optimal control. The derived algorithms successfully solved tabular hard exploration problems and out-performed the sample efficiency of SAC in various continuous control problems; thus closing the gap between what is conceivable and what is numerically feasible.

### BROADER IMPACT STATEMENT

Reinforcement learning provides a framework to solve complex tasks and learn sophisticated behaviors in simulated environments. However, its incapacity to deal with very sparse, deceptive and adversarial rewards, as well as its sample-inefficiency, prevent it from being widely applied in real-world scenarios. We believe investigating the classic optimality criteria is crucial for the deployment of RL. By introducing a novel family of optimality criteria that goes beyond exponentially discounted returns, we believe that we take a step towards more applicable RL. In that spirit, we also commit ourselves to releasing our code soon in order to allow the wider community to extend our work in the future.

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
