# OpenReview forum: "Delayed Geometric Discounts: An alternative criterion for Reinforcement Learning"
_ICLR.cc/2022/Conference — ICLR 2022 Submitted_

### Official Review · Reviewer_1aQD · 2021-11-01

**Correctness:** 3
**Technical Novelty And Significance:** 3
**Empirical Novelty And Significance:** 3
**Recommendation:** 5
**Confidence:** 3

**Main Review:**

**Evaluation.** I find the motivation of this paper to be compelling. Geometric discounting is of course standard in reinforcement learning, but there might be good reason to explore alternative kinds of temporal preferences. In this sense, I am sympathetic with the goals of the paper. I also find that the questions studied surrounding the new criterion are sensible: Is a contraction still guaranteed? How do RL algorithms handle this new kind of discounting?

I did find that a number of concepts were unclear, including the primary contribution of the work: Equation 1. I provide more details on what I found unclear below under "Primary Questions". Given the central role of Equation 1, I believe the paper should spend more time on making this concept _extremely_ clear up front. I did find Figure 1 quite helpful for this purpose. However, in the conclusion, the paper states " we propose a family of delayed discounted objective functions that captures a wide range of non-monotonic time-preference models." By the time I finished the paper I did not have a clean picture of what kinds of non-mononotic time-preferences can be captured by the new criterion. I believe adding further detail commenting on these other kinds of time-preferences would benefit the paper.

I also found the experimental results to be lacking in strength in at least one respect: Results in both Figure 4 and 5 are estimates of expected returns (I believe -- the y-axis label just says "Expected Returns") based on 25 samples. It is critical to include some measure of variance in the reported figures (ideally 95% confidence intervals). If the variance is too high so as to prevent us from drawing conclusions, I suggest removing the requirement that the initial state is sampled uniformly at random from all states, and instead choose a small handful of representative start states. This will considerably lower the variance.

The two new algorithms seem interesting, and I believe aspects of the new discounted criterion will be of interest to the community. I believe the presentation of this idea could be made more clean.

**Primary Questions**

- [PQ.1] I am having a hard time making sense of Equation 1, and specifically $\Phi_D(t)$. Since this is so central, I want to make sure I am understanding things correctly. First, what is $N$ in the superscript of the $\prod$ term? Is this a typo, and should be $D$? Second, the subscript of the sum seems odd: We require $(\sum_{d} a_d) = 1$, but $a_d$ takes values in $\mathbb{N}$ up to D. Consequently, for many values of $t$ that are not expressible as the sum of consecutive natural numbers, it is unclear what happens in Equation 1. For instance, let $t = 2$ and $D = 3$. Now, by my reading, the subscript of the sum enumerates over natural numbers including 0, 1, 2, and 3. However there is no sum of consecutive numbers such that their sum adds to 2. Do we not require that the $a_d$'s used be consecutive? Then there is ambiguity (between say, when $t=6$, the case where $a_d \in \\{2, 4\\}$ and $a_d \in \\{1, 5\\}$). If all this is trying to articulate is that we only use the first $t$ natural numbers, then the subscript of the sum should just read $\sum_{a_d=0}^{t-1}$

- [PQ.2] Two closely related papers come to mind that were not discussed. I believe the results are quite different, though the perspectives taken are similar. The first is "Rethinking the discount factor in reinforcement learning: A decision theoretic approach" by Pitis (AAAI 2019). This paper looks at what criteria are needed on the discount factor in order for the objectives of an MDP to cohere with the constraints of the axioms of classical decision theory. In fact, one of Pitis' results motivates the use of discounts that might place _more_ utility on future events, rather than the present. I believe the work by Pitis might give further motivation to the use of the proposed delayed discount offered here. Second, "Hyperbolic discounting and learning over multiple horizons" by Fedus et al. (RLDM/arxiv 2019) explore hyperbolic discounting. I believe some discussion of the differences between delayed discounting and hyperbolic might be warranted.


**Minor Questions**
- [MQ.1] Is gamma part of the _objective_, or an algorithmic tool? That is, are we defining the task we want the agent to solve, or, are we viewing choice of gammas an algorithmic solution to solve the true task? These are slightly different perspectives and I think it would be helpful to address which of these is intended early on. For instance, the paper states that the delayed criterion can benefit sample efficiency---this could read as though the true objective is unchanged, and that the agent is simply using a different discounting scheme as an algorithmic trick. In other words, I take this paper to be about rethinking discounting for RL, rather than on exploration. I found the related work section on exploration strategies out of theme for this reason.
- [MQ.2] "discount-based RL approaches are sample-inefficient and are slow at propagating interesting feed-backs to early states." I'm not sure I quite get this. As opposed to non-discount based RL?
- [MQ.3] Why is $Q^\pi$ also indexed by $\eta$ in Equation 3? I don't see $\eta$ appearing (is this the learning rate?) I see $\eta$ is again used to index $J^\pi$ in Equation 6 but $\eta$ does not appear anywhere on the right hand side of the equation. Are these only being used to indicate that we are considering the weighted objective from Equation 2? If so, there might be a better notation for this (or, I recommend calling specific attention to the meaning in the text).

**Writing Suggestions**
- You probably do not need the outline paragraph just before 2.1, and could remove it if space is tight.
- The plots are pixelated. I suggest generating pdf versions (doable in matplotlib with plt.savefig("*.pdf"))
- The font size in the axes and the legend of Figure 1 are far too small.
- Top of page 4: "in details" --> "in detail"
- "expectation over trajectories" of infinite length?
- "non stationarity horizon." --> "non-stationarity horizon."
- Typo: "a depth D higher than 6 7 the algorithm" --> This should either be just "7", or a hyphen is missing.
- "sum of the environments'" --> "sum of the environment's"
- I suggest not using $\mathcal{D}$ to represent a dataset of trajectories given the prevelance of $d$ and $D$ throughout the paper (with a different meaning).
- In Equation 5 and 6, it is stated that $s,a,s' \sim \mathcal{D}$ or $s \sim \mathcal{D}$ -- is it assumed there is some underlying probability distribution over the dataset that is being sampled from?
- "double Q-network trick" --> Consider citing the Double Q paper at this point.
- Algorithm 1, inputs, "pollyak" --> "Polyak"
- After Proposition 2: "and the execute the optimal policy" --> "and then execute the optimal policy"
- "whenever that it is diagonalizable" --> "whenever it is diagonalizable"
- "is it’s largest eigenvalue" --> "is its largest eigenvalue"
- Why are the subscript of the two expectations in Equation 10 different?
- What is $c(s,a)$ in Algorithm 2? I did not see it defined.
- "How does the different parameters" --> "How do the different parameters"
- "all possible states (uniform $p_0$) ." --> "all possible states (uniform $p_0$)."
- A \citet should be turned into a \cite: "exploration RL algorithmsEcoffet et al." --> ard exploration RL algorithms\cite{ecoffet...}""
-

**Summary Of The Paper:**

**Summary.**
This paper proposes a generalization to the typical geometric discounting scheme in reinforcement learning. In particular, a new _delayed_ discount criterion is proposed (Equation 1) that captures traditional geometric discounting when $D=0$, but also allows for other kinds of temporal preferences. Figure 2 provides a clean visual illustrating the different kinds of discounting afforded for different settings of $D$. Following this, several natural questions emerge: What happens to the $\gamma$-contraction in repeated application of the Bellman Operator? Proposition 1 addresses this question by stating that a $\gamma_D$-contraction is still guaranteed. How might typical reinforcement learning algorithms handle this new discounting scheme? The paper addresses this question by introducing two new algorithms. The first (Algorithm 1) is a generalization of Soft Actor Critic, while the second (Algorithm 2) is a policy-based algorithm that approximates the finite horizon problem. The first set of experiments focus on three maze problems. Estimates of the expected returns of learned stationary policies are reported for Algorithm 1 under four conditions as the depth $D$ is changed from 0 up to 8. The main finding is that as $D$, returns do tend to increase up to a certain point, at which point it drops (around when $D = 6$ or $7$). Further experiments inspect the performance of Algorithm 2 in a similar way (Figure 5), and contrast the performance of Algorithm 1 with typical SAC on continuous control games like Ant-v2. On the chosen four games (Figure 6), the average reward of GSAC improves over SAC slightly, though the variance is arguably slightly higher.

**Summary Of The Review:**

This paper is conceptually interesting but could be refined in its execution. A few small aspects of it are unclear (including, to me, the main definition: Equation 1). I believe the paper will be strengthened with a much more clear exposition of the central idea, and improved rigor in presentation of the empirical results. For these reasons I lean slightly toward rejection at the moment, but am willing to increase my score if the paper can be changed in these two ways. For clarity, they are: (1) Answer PQ.1 and provide further exposition in the paper to make the proposed criterion more clear, and (2) Include a variance measure in the first two sets of experiments, and hedge the statements made about the results in light of the observed variance.

---

### Official Review · Reviewer_tzWw · 2021-11-03

**Correctness:** 3
**Technical Novelty And Significance:** 3
**Empirical Novelty And Significance:** 2
**Recommendation:** 6
**Confidence:** 3

**Main Review:**

The problem considered in the paper---that of generalizing the optimality criterion in RL---strikes me as potentially interesting  and with potential to broaden the applicability of RL to other domains. It also follows recent work in off-policy TD-learning---see, for example, the references [a, b] below, which also consider time-dependent discounting although, it seems to me, in a more restricted setting than that considered in this paper.

This said, there are several aspects of the paper that are not very clear to me and/or which would benefit from added discussion. Specifically:

- The considered discounts $\Phi_D$ are introduced with no motivation whatsoever. Where does this particular definition come from?
Without any introduction, this definition seems completely arbitrary.

- Still on the point above, the definition in (1) is far from clear. Can you provide some intuition on what it represents? Perhaps defining $\Phi_D$ recursively (as done above) may be more easily parsed by the reader, and makes the subsequent derivations much more plausible upon a first read.

- Under what conditions is the loss $\mathcal{L}_D(\pi,r)$ in (1) well-defined (i.e., is the series convergent)?

- I don't understand what is depicted in Fig. 2. What is the normalized coefficients? Normalized over what? Time (the text in page 2 seems to suggest it)?

- In the related work section, some discussion of the aforementioned works would certainly be of interest (please see refs. [a, b] below and other references therein).

- As an overall comment on the results, it is somewhat strange that SAC and GSAC are compared in terms of "average reward", when neither of the two is optimizing the average reward. Also, although the paper states that GSAC shows better sample efficiency across all tasks, if we take into consideration the error margins, the difference seems to be significant only for the walker and half-cheetah. Or am I missing something? Also, it would be of interest if some discussion could be made regarding any qualitative differences between the corresponding optimal policies, in light of the motivation for considering the framework discussed in the paper.

_Minor aspects:_

- In the beginning of Section 4 the paper says "we evaluate the performance of the proposed algorithm." But, actually, the paper proposes two algorithms...

- Still in the experiments, in the "Optimal Control Approximation", the text refers to the "red, blue and green curves", but does not refer to a particular figure (which I guess is Fig. 5).


- In the appendix, the reward seems to be denoted as $c$, which suggests a cost. Although there is no technical problem in that, it adds somewhat of a "cognitive dissonance" and may be worth making consistent.

_Refs._

[a] H. van Hasselt and A. Mahmood and R. Sutton. Off-policy TD$(\lambda)$ with a true online equivalence. In _Proc. 30th Conf. Uncertainty in Artificial Intelligence_, pp. 330-339, 2014.

[b] H. Yu and A. Mahmood and R. Sutton. On generalized Bellman equations and temporal-difference learning. _J. Machine Learning Res._ 19:1-49, 2018

**Summary Of The Paper:**

The paper proposes an optimality criterion for RL that is generalizes the standard exponential discount commonly used in a large volume of RL research. In particular, the paper proposes substituting the standard discount term $\gamma^t$ by a generalized discount term $\Phi_D(t)$ built from a set of discounts $\gamma_0,\ldots,\gamma_D$. The discount terms $\Phi_D$ can be defined recursively as

- $\Phi_0(t)=\gamma_0^t$ (corresponding to the standard exponential discounting);
- $\Phi_D(t)=\sum_{k=0}^t\gamma_D^k\Phi_{D-1}(t-k)$

The consideration of non-uniform discounting implies that the resulting optimal policies may, in general, be non-stationary. Nevertheless, the paper introduces an equivalent to the $Q$-functions in the standard RL formulation and shows that these functions can be computed using standard dynamic programming. Letting

$Q_d^\pi(s,a)=E\left[\sum_{t=0}^\infty\Phi_d(t)r_t\mid s_0=s,a_0=a\right]$,

it follows that $Q_0^\pi$ corresponds to the standard $Q^\pi$ with discount factor $\gamma_0$, and each $Q_d^\pi$ can be computed from $Q_{d-1}^\pi$ using a dynamic programming operator $T_\pi^d$ that has $Q_d^\pi$ as its unique fixed point. These functions can also be computed using existing RL algorithms as part of a generalized policy iteration cycle that---upon convergence---may yield an "optimal stationary policy" (convergence is not guaranteed, though).

The paper also proposes an approximate policy that breaks down the value function for the problem into two terms---one that accounts for the rewards in the first $H$ steps and the other for the remaining steps. Under mild conditions on the discounts $\gamma_0,\ldots,\gamma_D$, the latter term can be approximated by the standard optimal value function (with standard exponential discounting) for large $H$, allowing for the computation of an approximate (non-stationary) policy.

**Summary Of The Review:**

Although the premise of the paper is interesting, and the proposed approach is novel, to the extent of my knowledge. However, the proposed framework is introduced in a rather arbitrary manner; it is not clear how it relates with some existing work; the results are hard to interpret.

---

### Official Review · Reviewer_7SFq · 2021-11-03

**Correctness:** 3
**Technical Novelty And Significance:** 3
**Empirical Novelty And Significance:** 3
**Recommendation:** 3
**Confidence:** 4

**Main Review:**

In reinforcement learning, it is very common to use variants of a discounting parameter to emphasize the importance of immediate rewards as opposed to rewards received in the future. This is both a) intuitively plausible and b) beneficial for theory (convergence guarantees for learning and planning usually hinge on the Bellman equation being a gamma-contraction). But, as the paper argues correctly, geometric discounting results in weights that get exponentially smaller as we increase the horizon. This ultimately results in policies that are myopic in nature regardless of how large the discount factor is. While the average reward could be thought of as the limiting case of gamma-discounted returns, in principle using gamma values extremely close to 1 cannot address the myopic-policy problem since even small amount of noise in the learning process can compensate for negligible improvements of the less myopic policies.

The solution that this paper advocates for is to use generalized geometric discounting scheme with two hyper-parameters (\gamma, D). Intuitively, the role of the D parameter could be thought of as sacrificing the short term reward for ~O(D) steps before the geometric discounting kicks in.  As a micro point, I really struggled with the notation used to introduce the idea. There is Q_{\mu} that is only used once, then there is Q_{d} and Q_{D}. In equation 1, there is regular N and then there is \mathbb{N}. In general I don't think the notation used here is the most careful one, but that's a minor issue that will hopefully be fixed.

Going to the core of the idea, while I think this is a step towards learning less myopic policies, I think the original problem with the geometric discounts is still present in the generalization advocated here. In particular, the introduction of the D parameter is merely going to delay when the exponential nature of discounting kicks in, and does a marginal step to really eradicate the problem. The benefit of adding D is also, in my opinion, overshadowed by the fact that the discounting is now non-monotonic. That is to say, depending upon the setting of D, we may put less weight behind rewards that are received earlier in the trajectory relative to some of the rewards received later on. This means in principle, actions with disastrous short-term effects may be preferred so long as they lead into better long term ramifications. This is an undesirable property whose presence is not justified here. A second weakness is that the optimal policy is now non-stationary. Intuitively, I think this is because it actually matters how far in the trajectory we reach a certain state. If we go back to a state after a while, we may have to take a different action because the follow up states may now have larger or smaller weight. Thus, it is debatable if the marginal improvement in delaying the exponential decay actually justifies now having to deal with these technical issues.

In equation (4), it is stated that Q_{D} is the fixed-point of a Bellman like equation. Notice that in this equation, the reward itself depends upon the Q_d of all smaller ds. So while this may be a contraction all given Q_{d}s, it is not clear to me how to design a Policy Evaluation-like algorithm to solve for Q_{D}. While I conjecture that such an algorithm exists, the paper jumps into the function approximation case before a thorough treatment of Policy Iteration in the tabular case. Showing there exists a sound and convergent Policy Iteration algorithm in the tabular case is really the minimum we should desire to have. Notice that my main concern is due to the fact that solving for the fixed-point of Q_{D} requires knowing the fixed point for all previous ds. This is a) computationally expensive, and b) non-trivial in a convergence sense. In particular, I am not sure if the obvious algorithm of solving for lower level ds is actually convergent absence an analysis.

While I appreciate experiments on Mujoco, I don't think these Mujoco domains are hard in terms of exploration. If the motivation of the paper is to do better on hard-to-explore problems, experiments need to support the thesis of the paper, and to demonstrate that the new algorithm is capable of doing well on actual hard-to-explore problems. Notice also that the value of D used is pretty small here (D=1), which really defeats the whole purpose of using the alternative discounting scheme.

I also think two important comparisons are missing here. One is the idea of logarithmic discounting [1], which starts with the exact same motivation but proposes to do Q updates in the logarithmic space. And the second one is the idea of hyperbolic discounting [2], which is also known to be cognitively more plausible. It is surprising that these two works are not even acknowledged, let alone compared with.

I finally like to highlight that the example provided in Figure 1 accentuates not just the weakness of geometric discounting but the weakness of any form of discounting. For any discounting scheme it is possible to construct an MDP for which learning in presence of discounting can yield suboptimal results. In that sense, only the average reward setting can truly hedge against this issue by getting rid of discounting altogether. So it would be nice to know what I am missing here.

[1] van Seijen and friends "Using a logarithmic mapping to enable lower discount factors in reinforcement learning"
[2] Fedus and friends "Hyperbolic discounting and learning over multiple horizons"

**Summary Of The Paper:**

This paper proposes a discounting framework that generalizes the standard geometrically discounted reinforcement learning. The paper shows that the generalization still lends itself to deriving Bellman-like equations, and that it can provide empirical benefits on Mujoco problems.

**Summary Of The Review:**

In light of the issues I outlined above, especially in terms of the absence of theory for the tabular case, lack of experiments on truly hard-to-explore domains, and missing existing work, I am voting for rejection. That said, I certainly acknowledge issues that arise when using geometric discounting and I encourage authors to further explore this idea. A better discounting scheme may in theory have a high impact, but the current manuscript is not demonstrating to me (at least yet) that this is a good alternative.

---

### Official Review · Reviewer_hhP8 · 2021-11-06

**Correctness:** 2
**Technical Novelty And Significance:** 2
**Empirical Novelty And Significance:** 1
**Recommendation:** 3
**Confidence:** 4

**Main Review:**

I'm recommending rejection at this time, due to the following concerns:

1. The arguments made about the algorithm aren't convincing. e.g., there's some explanation for why the algorithm would have improved sample efficiency, but the empirical evaluation cited for the explanation doesn't readily provide evidence that the explanation is correct.

2. Many key implementation details are missing. The current paper (and supplement), as presented, are missing some key pieces of information to a point where it's not clear what was actually done. e.g., in the environments of Figure 3, the later text implies that they were tabular, but what were the sizes of their state spaces? Given discrete time, such information is key in setting expectations under the specified discount factor. Where are the error bars in Figure 4? How many runs were performed in Figure 6, and what do the shaded regions represent?

3. The bulk of the results in Figure 6 don't appear statistically significant. The claim that GSAC "shows better sample efficiency across all tasks" doesn't seem supported here, and consequently puts into question the explanation for why GSAC should have better sample efficiency.

Along the lines of the above concerns, I have the following questions:

1. What is the motivation behind considering Lη in the first place? There was discussion about delayed discounting, and how Equation 1 provided temporal weightings which discarded short term information and emphasized a delayed geometric weighting. With seemingly no context, a linear combination was then considered and became the focus of the work.

2. For the tabular results, what was the rationale for running SAC/GSAC in these environments, as opposed to a simpler algorithm to demonstrate the proposed objective? Along this line, as opposed to the stochastic samples averaged over several runs in these environments, could the solutions not be computed directly?

3. Can the authors comment on why the proposed method is unstable at higher depths in the tabular results? Is it some fundamental limitation of the objective, or does there exist some configuration where it learns something sensible (e.g., lower learning rate and more time)?

4. Can the authors clarify the following: "This has the beneficial side-effect of enhancing sample efficiency as it helps the agent to rapidly back-propagate long-term feed-backs to early states. This reduces the time needed to distinguish good from bad behaviors, particularly in continuous settings where function approximation are typically used to learn policies." Section 4.2 doesn't really discuss it, but claims the empirical evaluation demonstrates improved sample efficiency, and restates that this explanation is the case. What aspect of the empirical evaluation provides evidence that this is the reason behind the improvements?

5. Regarding Figure 7 in the supplement, is this "critical frontier" something to be expected of the objective? That is, will there be such a frontier in every environment, or is it just an observation in a single environment? Can the authors further comment on why the area above the frontier is dramatically inconsistent? I think this is another figure which would have greatly benefitted from computing an exact (or near exact) solution in a small tabular environment, and not be generated from stochastic samples under an algorithm with many moving parts.

**Summary Of The Paper:**

This paper generalizes the discounted objective in RL with "delayed" discounted objectives. They analyze the proposed objective and how it may be used to approximate optimal control, and evaluate the proposed method empirically.

**Summary Of The Review:**

In light of the above concerns pertaining to the proposed method's motivation, unclear arguments, and empirical methodology, I am recommending rejection at this time. I'm willing to raise my score should my questions and concerns (and potential misunderstandings) be adequately addressed.

---

### Decision · Program_Chairs · 2022-01-20

**Decision:**

Reject

**Comment:**

Overall, the reviewers were insufficiently enthused by this paper.  There was no rebuttal, and the authors did not engage or answer questions raised.  I concur with the reviewers, and encourage the authors to carefully consider the provided feedback.